# Barriers to g*BRCA* Testing in High-Risk HER2-Negative Early Breast Cancer

**DOI:** 10.3390/jpm13081228

**Published:** 2023-08-03

**Authors:** Olivia Foroughi, Shaheen Madraswala, Jennifer Hayes, Kara Glover, Liam Lee, Moumita Chaki, Stella Redpath, Agnes Weixuan Yu, David Chiu, Kristen Garner Amanti, Gary Gustavsen

**Affiliations:** 1Health Advances LLC, Newton, MA 02466, USA; 2AstraZeneca, Gaithersburg, MD 20878, USA; 3Merck & Co., Inc., Rahway, NJ 07065, USA

**Keywords:** g*BRCA*, olaparib, early-stage, HR+/HER2−, breast cancer, high risk, genomic risk score

## Abstract

Despite the OlympiA trial demonstrating that early-stage, high-risk, HER2- germline *BRCA1* and *BRCA2* mutation (g*BRCAm*) positive breast cancer patients can benefit from PARPi in the adjuvant setting, the g*BRCA* testing rate in early-stage HR+/HER2− patients remains suboptimal compared to that in early-stage TNBC patients. To better understand the perceived barriers associated with g*BRCA* testing in HR+/HER2− disease, a quantitative survey was conducted across stakeholders (*n* = 430) including medical oncologists, surgeons, nurses, physician assistants, payers, and patients. This study revealed that while payers claim to cover g*BRCA* testing, poor clinician documentation and overutilization are key challenges. Therefore, payers place utilization management controls on g*BRCA* testing due to their impression that clinicians overtest. These controls have led to healthcare professionals experiencing payer pushback in the form of reimbursement limitations and denials. The perceived challenges to g*BRCA* testing stem from the lack of consensus dictating which patients are high risk and should be tested. While payers define high risk based on the CPS + EG score from the OlympiA trial, HCPs adopt a broader definition including genomic risk scores, lymph node involvement, and tumor grade and size. A dialogue to harmonize risk classification and testing eligibility across stakeholders is critical to address this disconnect and increase g*BRCA* testing in appropriate patients.

## 1. Introduction

For over two decades, genetic testing for women with an increased risk of breast cancer due to germline *BRCA1* and *BRCA2* mutations (g*BRCA*m) has allowed for better treatment planning, intervention, and long-term survival. The discovery of *BRCA1* and *BRCA2* pathogenic variants in women with breast cancer in the 1990s paved the way for the expansion of personalized cancer care [1]. By 1996, g*BRCA* screening became the first genetic test as a clinical service to determine cancer risk [2]. While 13% of women will develop breast cancer sometime during their lives, women with g*BRCA* mutations are even more likely to develop the disease [3,4]. Studies have demonstrated that an estimated 46% of women with a *BRCA1* mutation born before 1920 developed breast cancer by the age of 70, with rates rising up to 59% for women born after 1950, with some carriers having risks above 90% at the 5th and 95th percentiles [4]. The risk of breast cancer in *BRCA2* mutation carriers is comparatively lower, with an average cumulative risk of 39% for women born before 1920 and up to 51% for those born after 1950 [4]. For some women in the 5th and 95th percentile, this risk can rise to over 90% [4]. Despite the low initial uptake of g*BRCA* testing, over time it has become a commonly accepted diagnostic test to determine susceptibility to breast cancer [5]. By 2005, the U.S. Preventive Services Task Force fully recognized the benefits of testing and recommended it for women with a family or personal history of an increased risk of g*BRCA*-related cancers [1].

Beyond preventative screening, germline *BRCA* tests can inform surgical treatment planning after an initial breast cancer diagnosis. While g*BRCA* testing may be relevant for all breast cancer patients, g*BRCA* mutations are even more common among certain breast cancer subtypes [6]. Breast cancer includes four widely recognized subtypes: luminal A (hormone receptor positive/human epidermal growth factor receptor 2 negative, HR+/HER2−), luminal B (HR+/HER2+), HR−/HER2+, and triple negative (HR−/HER2−). Patients with triple-negative breast cancer (TNBC) have the highest prevalence of g*BRCA* mutations (4.44% and 2.05% for *BRCA1* and *BRCA2*, respectively) followed by those with hormone-receptor-negative disease (3% and 2.61% for *BRCA1* and *BRCA2,* respectively) and hormone-receptor-positive disease (0.40% and 1.09% for *BRCA1* and *BRCA2,* respectively) [6]. It is estimated that 1–7% of all women with breast cancer have a g*BRCA1* mutation and 1–3% have a g*BRCA2* mutation, and as a result, the American Society of Breast Surgeons (ASBrS) Consensus Guideline on Genetic Testing for Hereditary Breast Cancer recommends that genetic testing should be offered to all breast cancer patients (newly diagnosed or with a personal or family history) for *BRCA* mutations for treatment-planning purposes [7,8,9]. Recently, testing has been proven even more crucial for identifying patients with g*BRCA*m, as these patients may benefit from certain targeted therapies. For example, the inhibition of poly (ADP-ribose) polymerase (PARP) could lead to the formation of double-strand DNA breaks, which are lethal to the cell [10].

The OlympiA trial demonstrated that early-stage, high-risk, HER2-, g*BRCA*m-positive breast cancer patients can benefit from the PARP inhibitor, olaparib, in the adjuvant setting. The patients studied during the phase III, double-blind, randomized trial had HER2- breast cancer with g*BRCA1* or g*BRCA2* mutations and high-risk clinicopathological factors [11]. The patients were randomly assigned, in a 1:1 ratio, to one year of oral olaparib or placebo, with a primary end point of invasive disease-free survival and a key secondary endpoint of overall survival [11]. The study revealed that olaparib significantly improved overall survival (OS) in the patients with HER2-, high-risk, early-stage g*BRCAm* breast cancer and reduced the risk of death over placebo by 32% [11]. As a result of the OlympiA trial findings, in March of 2022, the United States Food and Drug Administration (FDA) updated the olaparib approval to include the treatment of adult patients with deleterious or suspected deleterious g*BRCA*m, HER2-negative, high-risk early breast cancer who have been treated with neoadjuvant or adjuvant chemotherapy [12]. Shortly after the U.S. FDA label expansion, the Committee for Medicinal Products for Human Use recommended the approval of olaparib for patients in the European Union (EU) as a monotherapy or in combination with endocrine therapy for the adjuvant treatment of adult patients with g*BRCA* mutations who have HER2- high-risk early breast cancer previously treated with neoadjuvant or adjuvant chemotherapy [13].

The OlympiA study defined high risk for HER2- early-stage breast cancer patients as four pathologically confirmed positive lymph nodes or a CPS + EG score of three or higher, in conjunction with adjuvant and neoadjuvant therapy, respectively [11]. The CPS + EG scoring system estimates the probability of tumor relapse based on the clinical and pathologic stage as well as estrogen receptor status and histologic grade [12]. In addition to the OlympiA-defined CPS + EG score and pathologically confirmed lymph nodes, clinicians may consider other prognostic factors to determine patient risk of tumor recurrence. Historically, clinicians have used a range of factors such as nodal involvement and tumor size as determinants of risk of recurrence, while newer predictive tools such as genomic signatures (i.e., Oncotype Dx and MammaPrint) have become more common today in clinical practice [14].

Due to the success of the OlympiA trial, the National Comprehensive Cancer Network (NCCN) guidelines have been updated to acknowledge the role of g*BRCA* as a predictive biomarker and to recommend g*BRCA* testing to aid in treatment decisions with olaparib for patients with early-stage, high-risk, HER2- breast cancer [15]. Furthermore, the American Society of Clinical Oncology (ASCO) released a rapid guideline recommendation update to support the use of adjuvant olaparib in patients with early-stage, high-risk HER2- breast cancer and a g*BRCA* mutation [16].

Regardless of both the clear benefits of g*BRCA* testing alongside clinical guidelines and professional society support, not all eligible patients are tested today, particularly HR+/HER2− patients. Compared to patients with TNBC who were also included in OlympiA, far fewer breast cancer patients with early-stage, high-risk, HR+/HER2−, and g*BRCAm* receive olaparib due to a lack of comprehensive testing [17]. Moreover, long-term survival for g*BRCA2*m patients with HR+/HER2− breast cancer is in fact worse than long-term survival for patients with HR−/HER2− disease [18]. It is therefore critical that patients with HR+/HER2− breast cancer receive g*BRCA* testing as part of their care and are offered olaparib if appropriate.

To understand the perceived challenges with g*BRCA* testing in the United States, a survey was conducted across stakeholders, including medical oncologists, surgeons, nurses, physician assistants, payers, and importantly, patients. This study is the first of its kind, to our knowledge, to review a wide range of stakeholders to discover disconnections that reduce patient access to testing and ultimately to potentially survival-boosting targeted therapy.

## 2. Materials and Methods

### Survey Methodology

We conducted an online quantitative survey on medical oncologists (*n* = 94), surgeons (*n* = 97), nurses and physician assistants (*n* = 58), payers (*n* = 40), and patients (*n* = 141) to inform our review of the breast-cancer-testing landscape. The respondents’ demographics are displayed in Table 1.

Interviews. We performed a series of 30–60 min double-blinded phone-based interviews with oncologists, surgeons, nurses, and payers to help develop our survey instrument (Figure 1).

Questionnaire. The questionnaire consisted of three separate online surveys: one for the healthcare professionals, including medical oncologists, surgeons, nurses, and physician assistants; one for the patients; and one for the payers. To ensure that we obtained high-quality data, the 430 survey respondents were recruited by market research vendors in compliance with industry standards in addition to outreach among a proprietary database of experts. All the respondents were 21 or older at the time of the survey. The IRB determined that the research project was exempt from IRB oversight in accordance with the Department of Health and Human Services regulations 45 CFR 46.104(d) (2), with a review and final approval achieved on the Advarra CIRBI Platform (Figure 1). All the survey respondents reported involvement in g*BRCA* testing either as a healthcare professional, a payer determining a medical policy or reimbursement, or a patient with early-stage breast cancer that was eligible for testing.

Patients. The respondents were required to have been diagnosed with early-stage HR+/HER2− breast cancer within the past three years. The respondents could not be self-insured or uninsured.

Oncologists, Surgeons, Nurses, and Physician Assistants. The respondents were required to have been in practice for 2 to 36 years, spend more than 20% of their time in direct patient care, and manage more than 10 breast cancer patients per month. More than 10% of their patients had to have early-stage breast cancer, and more than 5% of their patients had to have HR+/HER2− breast cancer.

Payers. The respondents were required to have the title of medical director, clinical advisor, laboratory benefits manager, or chief medical officer with experience working at payer organizations for >2 years at plans covering >10,000 lives. The respondents were required to be frequently or directly involved in medical policy and reimbursement decisions for diagnostic tests relating to oncology.

## 3. Results and Discussion

This study revealed several critical findings, including one plausible explanation for why healthcare professionals fail to test nearly one third of their eligible patients for a g*BRCA* mutation. Healthcare professionals and patients view payer policies and reimbursements as the primary obstacle for testing all eligible breast cancer patients today. The perceived challenges to g*BRCA* testing coverage, reimbursement, overutilization, and documentation are all a result of the lack of agreement on which patients have a high risk of recurrence and therefore should be eligible for g*BRCA* testing based on clinical guidelines. This study identified key disagreements in risk assessment and g*BRCA* testing eligibility across all stakeholders. With a better understanding of stakeholder viewpoints and a harmonized risk classification, we can support breast cancer patients’ access to g*BRCA* testing.

### 3.1. Challenges to gBRCA Testing

Based on the survey results, payers claim to widely cover g*BRCA* testing for early-stage, high-risk, HR+/HER2− breast cancer patients and raise concerns that clinicians fail to properly document patient need and will overorder testing for patients (Figure 2). Payers suggest that they will receive an order from a clinician for a g*BRCA* test without appropriate documentation to demonstrate patient need, such as for use as a companion diagnostic, ancestry associated with breast cancer related to g*BRCA*m, or a personal history of relevant cancer. Moreover, healthcare professionals admit to ordering similar rates of testing for low-risk and high-risk early-stage HR+/HER2− breast cancer patients despite the lack of full support for low-risk testing in the current clinical guidelines. It is therefore unsurprising that only 36% of the payers surveyed will rely on the prescribing physician’s risk assessment alone to determine patient coverage eligibility (Figure 3). The other 64% of payers report conducting independent assessments based on either NCCN guidelines or predetermined high-risk criteria to decide if a patient should receive coverage for a g*BRCA* test (Figure 3). One possible explanation is that payers place significant utilization management controls on g*BRCA* testing due to their impression that clinicians are too imprecise with their definition of high risk and their documentation of patient eligibility.

Medical oncologists, surgeons, physician assistants, and nurses all believe that limited payer coverage is the most challenging hurdle for g*BRCA* testing today among their early-stage, high-risk, HR+/HER2− breast cancer patients (Figure 2). This perception mainly stems from payers upholding different definitions of risk than some clinicians (Figure 4). While payers largely consider a CPS + EG score equal to or greater than three if treated with neoadjuvant chemotherapy to be the primary definition of risk, medical oncologists and surgeons use a broader definition (Figure 4). As a result, poor insurance coverage is followed closely by a lack of clear guidelines around patient eligibility and the definition of high risk across stakeholders (Figure 2). Payer controls, such as prior authorization and outright rejections, lead to healthcare professionals’ perceptions that some payers create a barrier to testing. Discussions with healthcare professionals revealed that if they anticipate payer challenges, they may be less inclined to order the g*BRCA* test for their patients, and thus patients will not be eligible for targeted therapy without the appropriate companion diagnostic.

Patients similarly perceive payer controls as their top obstacle to g*BRCA* testing (Figure 2). The patient respondents reported high out-of-pocket costs associated with g*BRCA* testing and genetic counseling, in addition to coverage for genetic counseling as their top three challenges (Figure 2). All the surveyed patients reported having early-stage, HR+/HER2− breast cancer, with the majority disclosing their high-risk status, and therefore they should have had coverage for their companion test based on potential eligibility for olaparib. Among the patients surveyed who did not receive a g*BRCA* test, 46% reported hearing from their healthcare professional that their individual level of risk did not require g*BRCA* testing, although 80% of these respondents attested that they were high risk and therefore should have been tested (Figure 5). The lack of clarity around the guidelines and payer coverage trickles down to breast cancer patients and highlights the true impact of the incohesive classification of risk.

### 3.2. High-Risk Status

The perception of limited payer coverage and reimbursement as the most significant challenges to testing accentuates the disconnect between healthcare professionals, patients, and payers. This study’s findings suggest that payers believe that healthcare professionals overutilize g*BRCA* tests for patients that are not clinically indicated based on poor clinician documentation of patient need. Healthcare professionals in turn view the controls as excessive and burdensome, which may reduce their motivation to test patients if they expect that they will ultimately be met with prior authorization or outright denial. Patients bear the ultimate burden of the disagreement around risk status, as insufficient risk was the top reason for not receiving a g*BRCA* test, according to the patients surveyed in this study.

Importantly, as demonstrated in this study, the perceived challenges to g*BRCA* testing coverage, reimbursement, overutilization, and documentation all stem from the lack of consensus and clear guidelines dictating which patients are considered to have a high risk of recurrence and therefore should be tested.

The OlympiA trial, which supported adjuvant olaparib for early-stage breast cancer patients, utilized a risk metric defined as patients having a CPS + EG score of greater or equal to three and four or more positive lymph nodes [11]. While payers align their definition of high risk with the OlympiA trial based on the CPS + EG score, healthcare professionals have adopted a broader definition including genomic risk scores, lymph node involvement, tumor grade, and tumor size (Figure 4). Confusion around the appropriate definition of risk is not a new problem within the breast cancer community. The definition of recurrence risk has spurred working groups such as the IRIDE (hIgh Risk Definition in breast cancer) to develop a synthesized list of high-risk definitions to support clinicians in patient management [14].

A dialogue around harmonized risk classification and testing eligibility across stakeholders will be critical to address this disconnect and increase g*BRCA* testing in the appropriate patients. It is incumbent upon all stakeholders to rally behind a consistent set of criteria for risk status and eligibility for g*BRCA* testing.

Of note, genomic risk signatures (e.g., Oncotype DX, MammaPrint) were among the top risk criteria for clinicians but were comparatively low on the list for payers (Figure 4). To better align stakeholders, additional work may be necessary to entrench genomic classifiers as a key component of risk and eligibility for g*BRCA* testing in order to ensure that more women have access to life-saving treatments in the adjuvant setting.

While secondary issues such as access to genetic counseling and turnaround time were noted among stakeholders, these pain points were not identified as significant barriers to g*BRCA* testing (Figure 2).

### 3.3. Future of gBRCA Testing

This study revealed that clinicians are testing 63–69% of their early-stage, high-risk, HR+/HER2− breast cancer patients, demonstrating similar rates to an electronic medical record (EMR)-based real-world-evidence (RWE) study (Table 2, 17). In the next five years, many stakeholders are optimistic about a reduction in barriers and expect to increase the current testing rates for early-stage, high-risk, HR+/HER2− breast cancer patients. They suggest that increases in testing will come along with anticipated enhanced clarity around the definition of high risk and improved payer testing coverage (Figure 6).

Refining clinical guidelines and payer policies will be crucial to ensure that eligible patients receive g*BRCA* testing. This study and others constitute a meaningful step towards generating a broader range of risk definitions for clinicians and payers. Payers in particular derive policies based on peer-reviewed journals and clinical guidelines, which could be a valuable source to bridge the gaps between stakeholders in the future (Figure 7). With the combination of NCCN guidelines, peer-reviewed journals, and discussions at tumor boards, stakeholders can better align on an agreed-upon definition of the factors describing patient risk of cancer recurrence.

Collectively, engaging all stakeholders around one harmonized definition of risk and test eligibility will address the core disconnect and thereby improve patient care.

### 3.4. Limitations

The findings of this study should be considered in light of a few potential limitations that may point towards topics to be addressed in future research.

This research relies upon responses to an online survey from a sample of medical oncologists, surgeons, nurses, physician assistants, payers, and patients. In order to ensure high-quality responses, the survey only admitted qualified respondents to complete the full survey based on their responses to the screening criteria, including years in practice and number of breast cancer patients seen per month for healthcare professionals and insurance, age, and time since breast cancer diagnosis for patients. Additionally, to comply with gift ban laws in individual states, no respondents from Maine or Vermont were permitted to take the survey. Due to these restrictions, the final sample size was relatively small compared to the full population size. This study did not capture the race or ethnicity of the nonpatient respondents and therefore it is not possible to ascertain sample demographics relative to the full medical oncologist, surgeon, nurse practitioner, physician assistant, and payer community. In addition, the data were not statistically tested to quantify the differences between the categories of the respondents. It is important to recognize that each stakeholder group may still not be wholly representative of the broader population and in fact may not be a truly random sample. The screening restrictions limited the acceptance of some respondents into the survey, which may have introduced bias into the results.

Unfortunately, due to sampling limitations, the patient responses are not fully representative of the broader United States breast cancer population. The sample underrepresents Latin American/Hispanic patients relative to the U.S. population, with only 3% of the patients identifying as Hispanic/Latin American compared to 19% of the U.S. population [20]. This study did not report notable differences between patients of different ethnic groups due to the limited sample size. However, future work to identify the specific challenges experienced by different ethnic groups would be a worthwhile endeavor and would add substantial value to the available academic literature, address challenges with health equity, and impact patient care.

Moreover, this study relies upon self-reported data that cannot be independently verified. Self-reported data may contain biases due to the potential for exaggeration as well as selective memory. For example, it is possible that patients are less familiar with the specifics of their care and may not fully recall why they were told by their doctor that they were not eligible for g*BRCA* testing, as reported in Figure 5. Therefore, there is a possibility that some responses may be inaccurate, and it is not possible to verify all survey responses.

Finally, this study exclusively relies upon data from respondents in the United States and may not be broadly applicable to all geographies. Future research utilizing quantitative studies outside of the U.S. would allow the breast cancer community to identify opportunities to reduce global barriers to g*BRCA* testing access for patients.

## Figures and Tables

**Figure 1 jpm-13-01228-f001:**
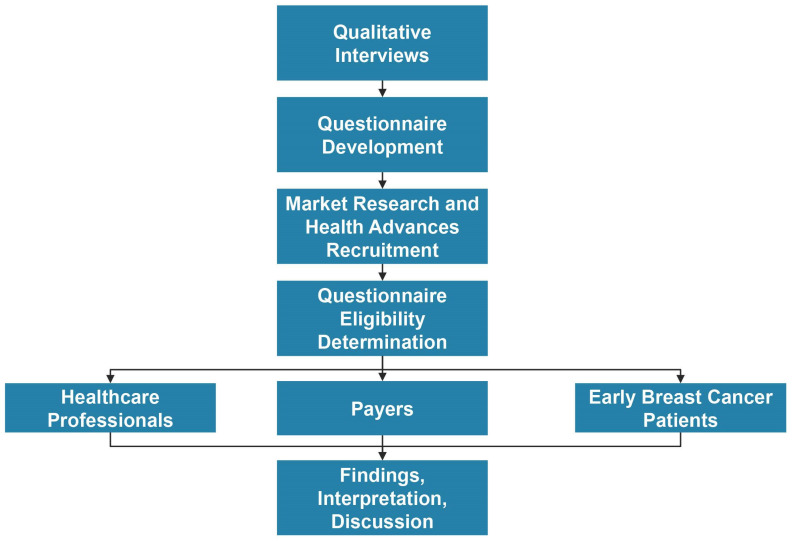
Study workflow. We began research by conducting qualitative interviews with experts to inform questionnaire development. We worked with market research firms and internally developed expert databases to recruit survey respondents. Survey respondents were either qualified and completed the survey or were deemed ineligible and not permitted to take the survey. We then reviewed findings, interpreted data, and discussed.

**Figure 2 jpm-13-01228-f002:**
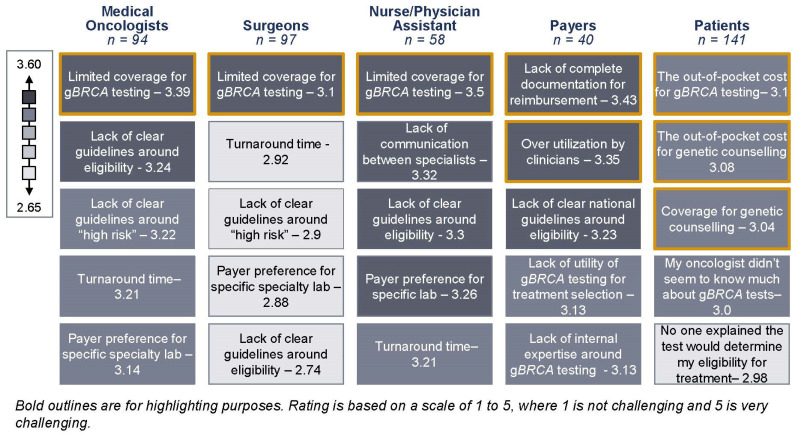
Top five reported barriers by healthcare professionals, payers, and patients. Outlines are for highlighting purposes only. Rating is based on a scale from 1 to 5. Medical oncologists (*n* = 94), surgeons (*n* = 97), and nurses and physician assistants (*n* = 58) were asked the following question: “Please rate how challenging the following factors are to g*BRCA* testing for early-stage, high risk, HR+/HER2− breast cancer patients today, on a scale of 1 to 5, with 1 being not challenging and 5 being very challenging.” Payers (*n* = 40) were asked the following question: “Please rate how challenging the following pathways/factors are to g*BRCA* testing for early-stage, HR+/HER2− breast cancer patients today, on a scale of 1 to 5, with 1 being not challenging and 5 being very challenging.” Patients (*n* = 141) were asked the following question: “Please rate the following challenges to accessing the *BRCA* testing process today, on a scale of 1 to 5, with 1 being not challenging and 5 being very challenging.” Only the top five barriers are included in the figure. Responses are listed in order of largest to smallest challenge and shaded grey.

**Figure 3 jpm-13-01228-f003:**
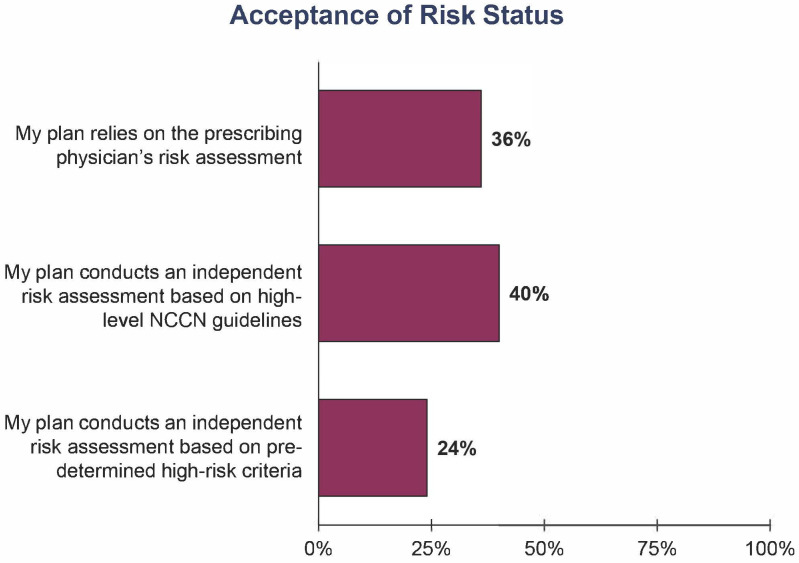
Payer acceptance of clinician-documented risk status. Payers (*n* = 40) were asked the following: “Which of the following best describes the process followed for assessing systemic recurrent risk status for early-stage, HR+/HER2− breast cancer patients covered by your plan?”.

**Figure 4 jpm-13-01228-f004:**
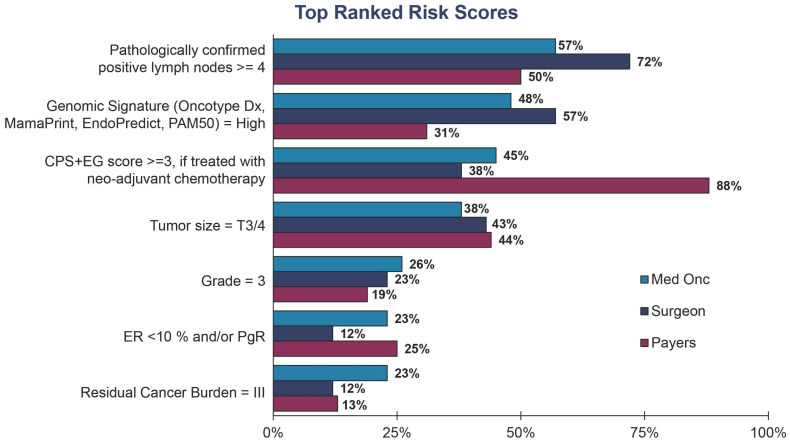
Stakeholders’ top three ranked risk factors for evaluating HR+/HER2−, early-stage breast cancer patients. Payers (*n* = 40) were asked the following: “Please rank, with the most important on top, the top 3 factors your plan considers when evaluating systemic recurrent risk status of early-stage, HR+/HER2− breast cancer patients?” Medical oncologists (*n* = 94) and surgeons (*n* = 97) were asked the following: “Please rank, with the most important on top, the top 3 factors you consider when evaluating recurrent risk status of early-stage, HR+/HER2− breast cancer patients under your care?”.

**Figure 5 jpm-13-01228-f005:**
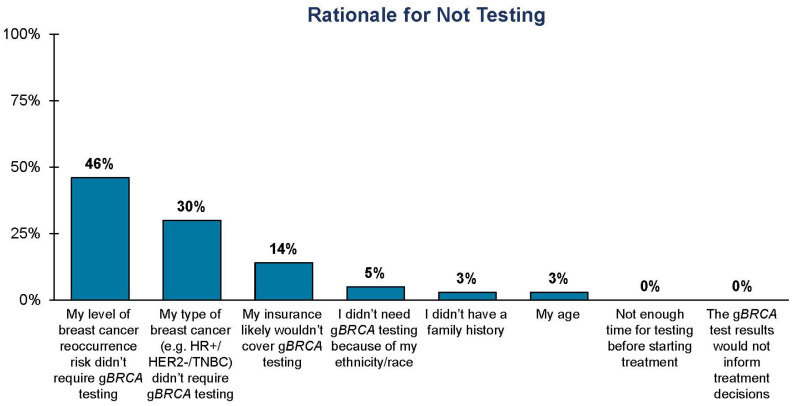
Patient-reported clinician rationale for not testing. Only patients who attested that they were told by their doctor that they did not need g*BRCA* testing were eligible to answer this question. Patients (*n* = 37) were asked the following: “Please rank, with the most important on top, the top three reasons why your doctor told you that you did not need *BRCA* testing.”.

**Figure 6 jpm-13-01228-f006:**
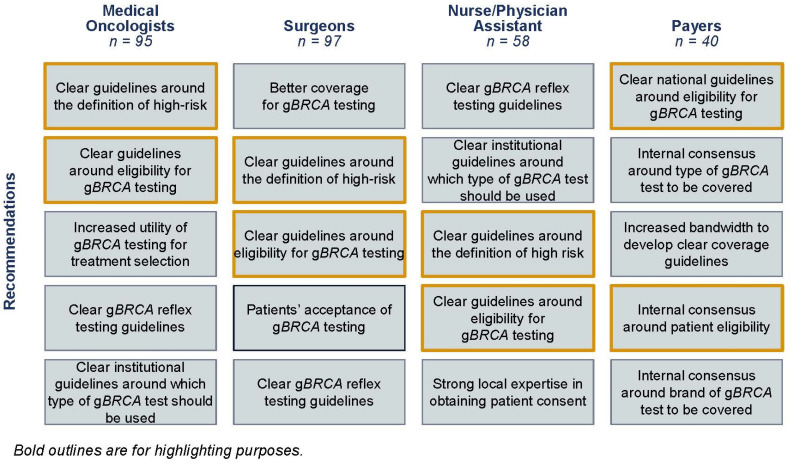
Drivers of increased testing in 5 years. Respondents were asked the following: “Which of the following factors do you expect will lead to an increase in g*BRCA* testing in early-stage, high-risk, HR+/HER2− breast cancer patients in 5 years? Please select all that apply. Which of the following factors do you expect will lead to a decrease in g*BRCA* testing in early-stage, high-risk, HR+/HER2− breast cancer patients in 5 years? Please select all that apply.”. Yellow border is used for highlighting purposes.

**Figure 7 jpm-13-01228-f007:**
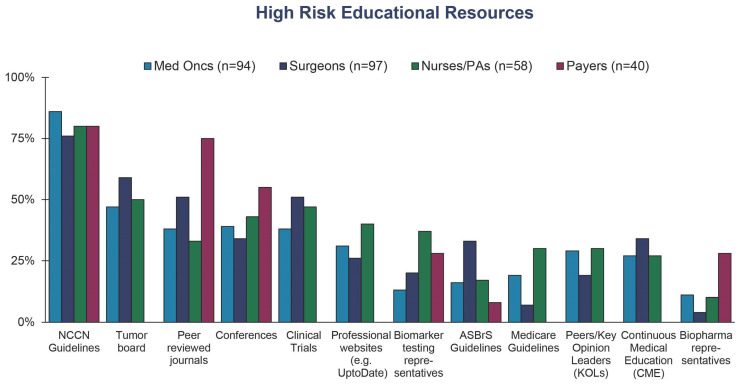
Resources for understanding risk of recurrence in breast cancer patients. Healthcare professionals were asked the following: “Which of the following resources do you use to inform your definition of high-risk for recurrence?” Payers were asked the following: “please select all that apply. Which of the following educational resources do you use to learn about g*BRCA* testing for breast cancer patients?”.

**Table 1 jpm-13-01228-t001:** Demographics of all survey respondents delineated by stakeholder type. Medical oncologists, surgeons, nurses, and physician assistants are sometimes referred to as “healthcare professionals” throughout the discussion. As of the time of publication, 23.6%, 20.6%, 38.6%, and 17.1% of the total United States population lives in the west, midwest, south, and northeast, respectively [19].

Oncologist Demographics (*n* = 94)			
	Average Number of Patients/Month	Average Low-Risk Testing Rate	Average High-Risk Testing Rate
Early-Stage Breast Cancer	88	44%	63%
		Respondents	
Geographic Region	West	20%	
	Midwest	14%	
	South	42%	
	Northeast	24%	
Practice Type	Private Practice	45%	
	Academic Health System	22%	
	Community Health System	33%	
Surgeon Demographics (*n* = 97)			
	Average Number of Patients/Month	Average Low-Risk Testing Rate	Average High-Risk Testing Rate
Early-Stage Breast Cancer	61	42%	66%
		Respondents	
Geographic Region	West	12%	
	Midwest	15%	
	South	36%	
	Northeast	36%	
Practice Type	Private Practice	34%	
	Academic Health System	31%	
	Community Health System	35%	
Nurse and Physician Assistant Demographics (*n* = 58)			
	Average Number of Patients/Month	Average Low-Risk Testing Rate	Average High-Risk Testing Rate
Early-Stage Breast Cancer	68	46%	69%
		Respondents	
Geographic Region	West	12%	
	Midwest	27%	
	South	44%	
	Northeast	17%	
Practice Type	Private Practice	35%	
	Academic Health System	12%	
	Community Health System	53%	
Payer Demographics (*n* = 40)			
		Respondents	
Geographic Region	West	30%	
	Midwest	20%	
	South	35%	
	Northeast	15%	
Average Plan Size (Lives)	10,000–100,000	10%	
	100,000–1,000,000	18%	
	1,000,000–5,000,000	43%	
	5,000,000–10,000,000	10%	
	>10,000,000	30%	
Patient Demographics (*n* = 141)			
		Respondents	
Sex	Female	100%	
Cancer Stage	Early Stage	100%	
Geographic Region	West	24%	
	Midwest	16%	
	South	38%	
	Northeast	23%	
Ethnicity	White	79%	
	Latin American/Hispanic	3%	
	Black/African American	17%	
	American Indian or Alaska Native	1%	
	Asian	1%	
Insurance Type	Private Insurance	80%	
	Medicare	6%	
	Medicaid	14%	

**Table 2 jpm-13-01228-t002:** Rates of g*BRCA* testing. Medical oncologists (*n* = 94), surgeons (*n* = 97), and nurses and physician assistants (*n* = 58) were asked the following: “What percentage of early-stage, HR+/HER2− breast cancer patients receive germline BRCA testing?”.

Respondent	Low Risk	High Risk
Medical Oncologist	44%	63%
Surgeon	42%	66%
Nurses and Physician Assistant	46%	69%

## Data Availability

Data are available from the corresponding author upon reasonable request.

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
