# Peer review of "Barriers to gBRCA Testing in High-Risk HER2-Negative Early Breast Cancer"

_jpm, 2023, doi:10.3390/jpm13081228_

Round 1

Reviewer 1 Report

The authors describe a survey of HCPs, payers and patients around testing ER+ HER2- breast cancers for BRCA1/2. They show that <70% of high risk patients are approved by insurers. Barriers appear to include lack of clear guidelines on testing and definition of high risk. The survey is interesting and important and highlights a need to improve guidance across the piece. The introduction needs substantial editing due to inaccurate quoting and referencing. The applicability of this paper is very US centric and that should be admitted as a limitation.

Specific comments

11.       Abstract: ‘A dialogue to harmonize risk classification and testing eligibility across stakeholders is critical to address ‘this’ disconnect and increase gBRCA testing in appropriate patients.’ -Suggest adding ‘this’ for clarity

22.       ‘The 1990s discovery of BRCA1 and BRCA2 ‘mutations’ in women with breast cancer paved the way for the expansion of personalized cancer care [1].’ I suggest changing ‘mutations’ to ‘pathogenic variants’ as ‘mutation is not now considered the best word in this context

33.       ‘While 13% of women will develop breast cancer sometime during their lives, 55-72% of women who inherit BRCA1 and 45-69% of women who inherit BRCA2 will develop breast cancer by ages 70 to 80 [3-5].’ -Please avoid using the Chen publication as it used modelling and included women from much earlier birth cohorts. The prospective references are much better but using the upper limit of 72% and 69% is misleading as this is an averaged risk. In reality some BRCA1/2 carriers have risks above 90% -see BOADICEA model

44.       ‘While gBRCA testing may be informative for all breast cancer patients, gBRCA mutations’ -This is debateable -no evidence that BRCA are more frequent than background for grade 1 if unselected for family history https://pubmed.ncbi.nlm.nih.gov/36442995/

55.       ‘Patients with triple negative breast cancer (TNBC) have the highest prevalence of gBRCA mutations (13.3%) followed by those with HR+/HER2- (5.9%) [7]’ The mean age in this reference of 8627 patients was, 50.5 years. This does NOT suggest this was an ‘unselected’ series. These rates from China do nor reflect large population based cohorts in Europe and North America. I would suggest using the CARRIERS study for a more realistic estimate from the relevant country. https://pubmed.ncbi.nlm.nih.gov/33471974/ This was only 1.49% for ER+ which would only drop slightly by excluding ER+ HER2+

66.       ‘. It is estimated that nearly 10% of all women with breast cancer have a gBRCAm, and as a result, the American Society of Breast Surgeons (ASBrS) Consensus Guideline on Genetic Testing for Hereditary Breast Cancer recommends that genetic testing should be offered to all breast cancer patients (newly diagnosed or with a personal history) for BRCA mutations for treatment planning purposes [8,9].’ This 10% figure is utter nonsense. Large population studies including CARRIERS and BRIDGES find only 2%. You cannot use this figure see also Dorling et al https://pubmed.ncbi.nlm.nih.gov/33471991/ Reference 8 states ‘However, it’s important to note that less than 10% of women diagnosed with breast cancer have a BRCA mutation.’ Ref 9 states ‘Approximately 10% of breast cancers are associated with a pathogenic germline variant in one of several different genes.3 More than 50% of pathogenic germline variants are mutations in the BRCA1 and BRCA2 genes.4-9’ -Thus even using ref 9 you should be using a MAXIMUM of 5% but even this is wrong!

77.       ‘The OlympiA trial demonstrated that early-stage, high-risk, HER2-, gBRCAm-positive breast cancer patients can benefit from the PARP inhibitor, Lynparza (olaparib), in the adjuvant setting’ -Should we only be using the generic name ‘olaparib’ here?

88.       ‘The CPS + EG scoring system estimates the probability of tumor relapse based on the clinical and pathologic stage as well as estrogen receptor stat(u)s and histologic grade [12]’

99.       ‘Furthermore, the American Society of Clinical Oncology (ASCO) released a rapid guideline recommendation update to support the use of adjuvant olaparib in patients with early-stage, high-risk HER2-breast cancer and a gBRCA mutation [14].’ -Please provide the FDA and EMA licensed indications for Olaparib in ER+ HER2- breast cancer.

110.   ‘Table 1. Includes the demographics of all survey respondents delineated by stakeholder types.’ Please provide proportion of US population in the designated areas

111.   The authors should cite that long term survival from ER= HER2- breast cancer is actually worse than ER- in BRCA carriers which further emphasises why high risk ER+ HER2- should be tested and offered Olaparib if appropriate https://pubmed.ncbi.nlm.nih.gov/33597716/

Author Response

Thank you very much for taking the time to review our manuscript “Barriers to gBRCA Testing in High-Risk HER2 Negative Early Breast Cancer”. We sincerely appreciate your expertise on this subject matter.

We have responded to the request to include alternative sources for citations in the introduction, including adding in all reviewer suggested sources. We have also corrected all clarifying recommended edits. Finally, we addressed the limitation that our paper is US centric in the limitations section.

Again, we want to reiterate that we appreciate the time and effort of the reviewer. We believe our study is impactful for the community to understand the current perceived barriers to gBRCA testing within breast cancer across multiple stakeholders.

Reviewer 2 Report

See enclosed comments.

See enclosed comments.

Author Response

Thank you very much for taking the time to review our manuscript “Barriers to gBRCA Testing in High-Risk HER2 Negative Early Breast Cancer”.

We have responded to all of the grammatical reviews. We have tried to respond to as many of the specific comments as possible, yet were not able to address all edits due to our conversation with our editor regarding the journal’s request for resubmission by July 25th.

Again, we want to reiterate that we appreciate the time and effort of the reviewer. We believe our study is impactful for the community to understand the current perceived barriers to gBRCA testing within breast cancer across multiple stakeholders.

Reviewer 3 Report

1Please discuss:

    The representativeness of the sample size and demographics.

2 Whether there are any notable differences among the patients of different ethnic groups.

          Numerous grammar issues have been identified. Please proceed with the necessary corrections and revisions to enhance clarity and readability. For instance, in the abstract section:

a.       Line 14: "gBRCA testing in HR+ HER2- disease" should be "gBRCA testing in HR+/HER2- disease."

b.       Line 18: "impression that clinicians over test" should be "impression that clinicians overtest."

c.       Line 21-22: "Payers define high-risk based on the OlympiA trial based on CPS + EG score" should be "Payers define high-risk based on the CPS + EG score from the OlympiA trial." 

Please thoroughly review the entire paper for any potential grammar issues.

Author Response

Thank you very much for taking the time to review our manuscript “Barriers to gBRCA Testing in High-Risk HER2 Negative Early Breast Cancer”.

We have responded to the Reviewer’s suggestion to discuss the representativeness of the sample size and demographics and notable differences among the patients of different ethnic groups in the limitations section. Additionally, we performed a thorough review to correct any potential grammar issues.

Again, we want to reiterate that we appreciate the time and effort of the reviewer. We believe our study is impactful for the community to understand the current perceived barriers to gBRCA testing within breast cancer across multiple stakeholders.